# Healthcare Supply Chain Management under COVID-19 Settings: The Existing Practices in Hong Kong and the United States

**DOI:** 10.3390/healthcare10081549

**Published:** 2022-08-16

**Authors:** Yui-Yip Lau, Maxim A. Dulebenets, Ho-Tung Yip, Yuk-Ming Tang

**Affiliations:** 1Division of Business and Hospitality Management, College of Professional and Continuing Education, The Hong Kong Polytechnic University, Hong Kong, China; 2Department of Civil & Environmental Engineering, College of Engineering, Florida A&M University-Florida State University (FAMU-FSU), Tallahassee, FL 32310-6046, USA; 3The Jockey Club School of Public Health and Primary Care, The Chinese University of Hong Kong, Hong Kong, China; 4Department of Industrial and Systems Engineering, The Hong Kong Polytechnic University, Hong Kong, China

**Keywords:** COVID-19, healthcare facilities, healthcare supply chain management, public health crisis

## Abstract

COVID-19 is recognized as an infectious disease generated by serious acute respiratory syndrome coronavirus 2. COVID-19 has rapidly spread all over the world within a short time period. Due to the coronavirus pandemic transmitting quickly worldwide, the impact on global healthcare systems and healthcare supply chain management has been profound. The COVID-19 outbreak has seriously influenced the routine and daily operations of healthcare facilities and the entire healthcare supply chain management and has brough about a public health crisis. As making sure the availability of healthcare facilities during COVID-19 is crucial, the debate on how to take resilience actions for sustaining healthcare supply chain management has gained new momentum. Apart from the logistics of handling human remains in some countries, supplies within the communities are urgently needed for emergency response. This study focuses on a comprehensive evaluation of the current practices of healthcare supply chain management in Hong Kong and the United States under COVID-19 settings. A wide range of different aspects associated with healthcare supply chain operations are considered, including the best practices for using respirators, transport of life-saving medical supplies, contingency healthcare strategies, blood distribution, and best practices for using disinfectants, as well as human remains handling and logistics. The outcomes of the conducted research identify the existing healthcare supply chain trends in two major Eastern and Western regions of the world, Hong Kong and the United States, and determine the key challenges and propose some strategies that can improve the effectiveness of healthcare supply chain management under COVID-19 settings. The study highlights how to build resilient healthcare supply chain management preparedness for future emergencies.

## 1. Introduction

A wave of severe acute typical respiratory diseases occurred in Wuhan, China, in December 2019. In fact, this was an undetermined viral disease before December 2019. Dry cough, tiredness, and fever were the most common symptoms of this virus [1]. Initially, the outbreak of severe acute respiratory syndrome coronavirus 2 (SARS-CoV-2) was recognized as a zoonotic transmission coming from a seafood market in Wuhan, China. Human-to-human transmission substantially intensified the virus spread. The disease generated by this virus is generally referred to as coronavirus disease 2019 (COVID-19) [2]. Italy was the first European country to fall under the impact of COVID-19 after the early transmission in China [3]. Indeed, the World Health Organization (WHO) quickly declared COVID-19 a global pandemic, which is described as “the occurrence of disease cases in excess of normal expectancy” [4]. COVID-19 has rapidly spread all over the world within a short time period because neither specific vaccines nor antiviral treatments were available. The virus quickly spread to over 118,000 cases and induced 4291 deaths within 114 countries [5]. Currently, this virus has been discovered in almost 200 countries and territories, crossing the entirety of six WHO regions (i.e., Europe, Eastern Mediterranean, Americas, South-East Asia, Africa, and the Western Pacific).

As of the beginning of June 2021, nearly 170 million confirmed cases and more than 3.6 million deaths were recorded due to COVID-19 [6]. A rapidly growing number of COVID-19 cases in Asia, the United States, and European regions led to the implementation of various strategies by the relevant authorities, such as “lock-downs”, “staying-at-home” orders, quarantines, social distancing policy, restrictions on public gatherings, public health campaigns (e.g., decreasing face touching, face coverings, and wearing masks in the public areas, decreasing the use of shared public spaces, and increasing handwashing), provisional shutdown of certain businesses (e.g., entertainment facilities, shopping malls, and restaurants) [7,8,9]. Online teaching and learning were implemented in response to “school-closures” globally [4,10,11]. To begin with, the number of COVID-19 cases was mostly observed among the older population groups (i.e., >65 years). Later, more and more cases were found in different age groups and specific population groups (e.g., children, pregnant women). According to the United Nations (2020) [12], people with fundamental medical conditions, such as heart problems, high blood pressure, and diabetes, or those of higher age, were suffering more severely from the negative impacts of COVID-19. Worse still, COVID-19 evolved quickly. New variants (e.g., Lineage B.1.1.7, N501Y) have been recognized in different parts of the world. As such, the new variants facilitated the rate of transmission and the viral load. The existing measures seem questionable in response to mitigating the further spread of the virus.

This virus, along with its variants, diffuses quickly and significantly increases the number of infected people in many geographical locations. In cases where communities are confronted with a lack of healthcare equipment and supplies (e.g., gloves, masks, coronavirus testing kits), a large number of infected people can cause an irrecoverable and unimaginable disaster [1]. Waves of COVID-19 infections are expected in the forthcoming years, along with enduring disturbance of worldwide healthcare supply chain provision [13]. Therefore, there is a need to conduct a comprehensive review of the existing healthcare supply chain management trends under COVID-19 settings, which will be the main focus of this study. In particular, this study performs a comprehensive evaluation of the current practices of healthcare supply chain management in Hong Kong and the United States under COVID-19 settings. Based on the recognized trends, it should be possible to provide useful insights and strategies for the relevant stakeholders to improve healthcare supply chain management. Along with the existing healthcare methods, a set of alternative approaches (e.g., approaches based on nature-inspired computing, artificial intelligence, and machine learning) should be considered, so that healthcare supply chain managers can effectively respond to the challenges imposed by COVID-19. Such innovative approaches could assist with a variety of complex tasks that include, but are not limited to, prediction of the COVID-19 spread, scheduling logistics operations (e.g., transportation of different medical supplies between hospitals serving COVID-19 patients), testing of the population for COVID-19, and evaluation of the effectiveness of different vaccines against COVID-19.

This paper is divided into six main sections. In Section 1, we discuss the relevant background information, COVID-19 trends, the concept of healthcare supply chain management, and the challenges of healthcare supply chain management in COVID-19 settings. In Section 2, we conduct a literature review relevant to the characteristics of healthcare systems, and the concepts and features of healthcare supply chain management in the context of the COVID-19 pandemic. In Section 3, we describe a case study approach as the main methodology in our research study. In Section 4, we provide detailed case studies of healthcare supply chain management in Hong Kong and the United States, two major Eastern and Western regions of the world. The uniqueness of healthcare supply chain management in Hong Kong and the United States is also reviewed and elaborated. Based on the five main areas (i.e., use of respirators, life-saving medical supplies, blood supply chains, disinfectants, and human remains handling and logistics), in Section 5 we investigate the key trends and recommend appropriate strategies for the related stakeholders to improve healthcare supply chain management in Hong Kong and the United States under COVID-19 settings. Concluding remarks, research limitations, and future research directions are given in Section 6.

## 2. Literature Review

The Supply chain process is the vital link in the delivery of healthcare services. The supply chain is a life-saving, crucial factor fostering the delivery of services and goods to the end-user, which in healthcare is the patient [13]. To a certain extent, managing the healthcare supply chain is difficult due to healthcare systems becoming more uncertain and sophisticated. Each step of the healthcare decision-making process includes a component of uncertainty and risks. Inefficient healthcare supply chain management may induce negative impacts on the healthcare system [14].

The outbreak of COVID-19 created one of the most vital disruptions of the past 50 years, “breaking many global supply chains” [4]. Healthcare supply chain operations have been substantially affected by COVID-19 and may face a serious lack of medical and healthcare equipment throughout the supply chain [15]. A number of major healthcare facilities (e.g., distribution centers, factories, and warehouses) are currently situated in COVID-19’s quarantine areas in China, Italy, India, the United States, Japan, Russia, and Australia [1,4]. Restricted access to personal protective equipment could increase the risk of infection for healthcare workers, providing implications for safety and health [13]. Furthermore, different healthcare supplies are sourced mostly from the quarantined regions. COVID-19 generates simultaneous supply and demand in a chaotic manner [4]. In the COVID-19 context, people are not allowed to travel outside their home country or city. As expected, they have to purchase medical care products to respond to the COVID-19 threat and paradigm changes [16] and begin to purchase and store medical care products. The border closures, plus the unpredicted changes in demand and supply, produced remarkable healthcare supply chain disturbances across the globe [9]. In such cases, effective healthcare supply chain management is crucial, as we need to make sure that necessary healthcare products are distributed diversely in a healthcare system at the right quality and quantity, and in time [17]. In response, building a resilient healthcare supply chain network that can control the flow between members of the healthcare supply chain, prepare for unforeseeable risk incidents, mitigate the disruptions, and oversee the inventory of medical products is vital. Additionally, determining the optimal location of warehouses and distribution centers, determining the optimal number of used vehicles, and allocating medical products to pharmacies and hospitals are crucial during the crisis of the COVID-19 pandemic [18]. It may decrease the waiting time of patients and increase resource distribution in the emergency department of hospitals under a COVID-19 environment to boost efficiency [19]. Throughout healthcare supply chain management, the relevant decision makers have to determine the required quantity of drugs (e.g., chloroquine phosphate, tocilizumab, interpherone alpha 2b, umifenovir, atazanavir, ribavirin, favipiravir, remdesivir, lopinavir, and hydroxy chloroquine) [18], oxygen, ventilation systems, masks, disinfectants, and other medical supplies that will be distributed among the high-priority locations [20].

The authors of [1] point out that the healthcare industry operates its supply chains differently when compared to other industries. In the healthcare industry, manufacturers, wholesalers, distributors, and service providers operate separately. In other words, there are different parties involved, resulting in a lack of information or knowledge sharing, collective efforts, and strategic alliance configuration. As a result, data inconsistency, inefficient business processes, and fragmented systems have arisen in the existing healthcare supply chains; ref. [21] also highlighted that healthcare supply chains have performed insufficient technological adoption and not included standard product identification numbers. Indeed, the key global challenges in healthcare supply chains can be addressed by recognizing the appropriate approaches, techniques, and technologies to foster healthcare service delivery in the world [17]. Such challenges hinder the healthcare industry in the development of effective and innovative supply chain strategies in response to the COVID-19 pandemic. At present, many communities are facing unresolved issues (e.g., insufficient healthcare equipment and supplies, lack of reliable blood supply chains, and lack of effective human remain logistics) that prevent them from dealing with the virus successfully. In this regard, Moosavi et al. (2022) [9] emphasized that big data analytics enhance the healthcare supply chain in three main dimensions, namely inter-functional integration, hospital-supplier integration, and hospital-patient integration. Specifically, data-driven healthcare supply chains have overcome the constraints of developing low-income country healthcare systems and improved operational flexibility to address the disruptions created by COVID-19.

## 3. Methodology

### Case Study

The case study approach is commonly adopted across multiple areas and disciplines. As such, a case study is a perfect methodology when a thorough and holistic investigation is required for a specific phenomenon [22]. In order to carry out an in-depth research study, we use a case study design to draw conclusions from comparative cases. In our research study, comparative case designs are feasible because similar situations have been observed under various scenarios and sophistication; [23] pointed out that “the case study approach is a useful method for examining phenomena still unexplored. Case studies allow the investigation of phenomena separately from their context examining specific variables”. Each individual case study still needs to be comprehensively conducted as some phenomena may face inaccessible problems, but the assembly of different case studies aims to foster the process of problem-solving and confirms the results [22]. Hence, we can generate a common explanation or synthesis across the cases. This principle is named replication logic [24].

Accordingly, we use two representative cases (i.e., Hong Kong and the United States) to explore specific characteristics and common healthcare supply chain management trends in COVID-19 settings. Hong Kong can be viewed as a representative country of Eastern regions of the world, and it is located close to Wuhan, China, where the pandemic started. On the other hand, the United States is one of the largest countries in the Western regions of the world, which experienced the largest number of COVID-19 cases and deaths as of June 2021. A detailed review of healthcare supply chain management under COVID-19 settings in Hong Kong and the United States will be conducted, focusing on some of the aspects that were found to be the most critical for both countries throughout the response to the pandemic, including the following: (a) use of respirators; (b) life-saving medical supplies; (c) blood supply chains; (d) disinfectants; and (e) human remains handling and logistics.

The main steps of the adopted research methodology are summarized in Figure 1. In the first step, the relevant data will be collected using appropriate data sources that are open to the public. In the second step, the collected data will be analyzed and synthesized for Hong Kong and the United States, focusing on the major healthcare supply chain management aspects identified above. After the data analysis, the main trends and common challenges in healthcare supply chain management under COVID-19 settings will be identified in steps 3 and 4, respectively. Potential resolution strategies to the identified challenges will be proposed and thoroughly discussed in step 5.

## 4. Results

### 4.1. Review of Healthcare Supply Chain Management in Hong Kong

Hong Kong is located in the Pearl River Delta, facing the South China Sea, and has an area of around 1104 km^2^ comprising Hong Kong Island, Kowloon Peninsula, the New Territories, Lantau Island, and 260 other islands [25]. Although the area in Hong Kong is relatively small compared with other countries, there are 43 public hospitals and 12 private hospitals [26]. The number of hospitals is even more than in many developed countries [27]. Similar to many other countries such as Japan, Hong Kong has recently been facing serious aging problems and rising operation costs. The healthcare industry is facing challenges in reinforcing patient safety and efficiently optimizing the supply chain. Due to a rapid increase in the demand for healthcare services, healthcare systems in Hong Kong are introducing more complex and thorough services.

In Hong Kong, procurement and supply chain management is mainly conducted by the Hospital Authority (hereafter called “HA”). According to a recent report from the HA [28], the total purchase expenditure in 2020 was over USD 18 billion, which covers drugs, equipment, medical consumables, repair and maintenance, administration and services, etc. As for the supply chain processes, there are three issues mainly considered in supply chain management in order to comply with international standards and regulations: efficiency, security, and traceability. The framework of the supply chain management in the HA is divided into the head office level and the cluster level, which comprises more than 400 staff members. Their major functions are management, procurement policy and strategy, stakeholder engagement, and information platform, etc. To facilitate supply chain management, several practices are adopted in the HA, including Product Classification and Codification (PCC), Enterprise Resource Planning System (ERPS), and Product Tracking and Tracing (PTNT) [29].

On 25 January 2020, the HA announced an immediate activation of the emergency response level in public hospitals to comply with the government decision to raise the response level from “Serious” to “Emergency” due to the COVID-19 spread [29]. The following sections of the manuscript provide a detailed review of healthcare supply chain management in Hong Kong, focusing on the use of respirators, life-saving medical supplies, blood supply chains, disinfectants, and human remains handling and logistics.

#### 4.1.1. Use of Respirators

In order to respond to COVID-19, the HA in Hong Kong set up hospital infection control teams to revisit and offer the N95 respirator fit test program for all healthcare workers, particularly for those who work in high-risk areas. As of March 2020, the protective gear in stock in Hong Kong hospitals for medical staff could only last for two months, with about 24 million surgical masks and 1.1 million N95 respirators available [30]. Hong Kong HA has issued new guidelines on mask usage amid the citywide shortage, following the standards of the Center for Health Protection (CHP) of Hong Kong. Staff were encouraged to change protective gear, such as N95 respirators, less frequently. This action was made to ensure that there were enough N95 respirators for frontline health care professionals [31].

#### 4.1.2. Life-Saving Medical Supplies

Starting from the first wave of COVID-19, personal protective equipment (PPE) (e.g., disposable gloves, face shields, eye protection, isolation gowns, caps, and surgical masks) was identified as a life-saving preventative tool in response to COVID-19. PPE has also been used for aerosol-generating procedures (AGPs) [32]. The disturbance of the global supply chain, the supply shortage of raw materials, and the high uncertain delivery schedule further led the HA to face a shortage of life-saving medical supplies. As of August 2020, Hong Kong hospitals received around 2900 cubic meters of medical supplies and PPE. In order to provide emergency response, the HA collaborated with Cathay Pacific Cargo to provide a fast and flexible delivery service in response to COVID-19 [33]. In addition, Hong Kong imported medical oxygen, mainly from China (USD 1294.84 K, 8,765,450 m^3^), Germany (USD 220.95 K, 1,495,740 m^3^), the United States (USD 10.95 K, 74,114 m^3^), the United Kingdom (USD 6.35 K, 43,012 m^3^), and Malaysia (USD 3.67 K, 24,824 m^3^) [34].

#### 4.1.3. Blood Supply Chains

The blood supply chain is determined by elastic and fixed demand, together with the availability of personnel. As such, researchers have commonly adopted stochastic mathematical and optimization models to manage uncertainty. Additionally, the optimum temperature for delivering plasma products is −20 °C or cooler. To maintain this temperature during transportation, suitable refrigerator-equipped vehicles are needed for transportation. Indeed, comprehensive training for personnel to perform plasma transportation is also required. They need to be trained on how to pack and carry and how to encounter numerous unforeseeable incidents, such as cold store breakdowns. Thus, transportation capacity is restricted because of the limited number of trained personnel and the appropriate cold refrigerators during the time of the COVID-19 pandemic. Blood has an expiration date and there was a low blood supply during the COVID-19 pandemic. Thus, a First In, First Out (FIFO) approach may help to decrease the disposal level and mitigate the public health crisis [35]. The expiration of blood will bring about a rise in the cost of blood transfusions [14]. Due to the lockdown of blood donation centers, social distancing measures, and the lack of confidence and fear of getting the infection during blood donation for the general public, blood donation has been significantly affected. Measures to restore and gain public and donor confidence in blood donation are extremely important. Since ensuring that safety is always the highest priority, the HA Blood Transfusion Service (BTS) Expert Panel on Blood and Blood Products Safety has carefully reviewed and made the appropriate modifications in blood donation arrangements under the threat of COVID-19. Several main enhanced measures in blood donation were undertaken [36]:Members of the public who have recently visited a particular place that was later found to have active community transmission of COVID-19 should be deferred from blood donation for at least 28 days from the date of departure.Members of the public who had close contact with a person positively tested for COVID-19 while that person was symptomatic should be deferred from blood donation for at least 28 days.Members of the public who tested positive for COVID-19 will be deferred from blood donation for at least 180 days after complete recovery.

#### 4.1.4. Disinfectants

Medical schools identified numerous disinfectants against coronaviruses. Based on the concentration level and the duration of exposure, some disinfectants are not appropriate for the skin and may induce either pain or irritation. As such, ethanol, isopropanol, benzalkonium chloride, and povidone-iodine are suitable for the skin, while bleach, hydrogen peroxide, and chloroxylenol are fit for surfaces [37]. Ref. [38] pointed out that the residents who buy disinfectants need to (1) concentrate on the detailed product description or information exhibited on the labels, such as the validity period and manufacturing date; (2) buy disinfectant products of reputable brands from renowned retailers. In general, the HA provides health advice for the public to keep good environmental hygiene, such as having disinfectant wipes for disinfecting tables prior to and after every use. In the case of areas that are contaminated by respiratory secretions, it is required to disinfect the neighboring area and surface with a suitable disinfectant (e.g., diluted household bleach, alcohol, etc.) [39]. Currently, hotels, homes, hospitals, offices, restaurants, and schools are eager to use professional disinfection services. The disinfection service providers need to be proven by the Environmental Protection Department (EPD) of Hong Kong [40] and the CHP. Indeed, the Hong Kong Sports Institute plans to collaborate with Chiaphua Industries Limited and the Hong Kong University of Science and Technology to produce a locally developed disinfectant, so as to keep their coaches and athletes safe during the COVID-19 pandemic and prepare for the Tokyo 2020 Olympic Games [41].

#### 4.1.5. Human Remains Handling and Logistics

In Hong Kong, a number of precautions were introduced for human remains handling and disposal under the COVID-19 settings according to the Category 2 level [42]. Suitable resuscitation rooms with full PPE, including N95 respirators, eye protection, fluid/water-resistant gowns, and disposable gloves, are required. The human remains should be placed in a leak-proof transparent and robust plastic bag not less than 150 μm thick and zippered closed. Pins are prohibited. The outside of the body bag must be carefully wiped in 1-in-4 diluted household bleach and allowed to air dry. Furthermore, the body bag needs to attach a yellow label/tag for identification. Because the handling of dead bodies should be performed under Category 2, embalming is prohibited while cremation is allowed [43].

### 4.2. Review of Healthcare Supply Chain Management in the United States

The American Hospital Association (AHA) performs an annual survey of hospitals across the United States for the entire healthcare system. Based on the most recent AHA survey, the United States has a total of 6090 hospitals with a total capacity of 919,559 beds [44]. The majority of community hospitals are located in Texas, California, and Florida (a total of 512, 359, and 212 hospitals, respectively). On the other hand, Rhode Island, the District of Columbia, and Delaware have the lowest number of community hospitals (a total of 11, 10, and 7 hospitals, respectively). The AHA provides a comprehensive COVID-19 bed occupancy tool that allows projecting the expected hospital bed occupancy across the entire country by state. Such a tool is able to assist the relevant stakeholders to determine the hospitals that operate at the capacity level or close to the capacity level under COVID-19 settings.

A detailed review of healthcare supply chain management in the United States under COVID-19 settings was conducted using information provided by the Association for Health Care Resource and Materials Management (AHRMM) of the American Hospital Association. The AHRMM is represented by a leading group of healthcare supply chain professionals in the United States. Moreover, the information reported by other agencies (e.g., Center for Disease Control and Prevention, Food and Drug Administration, Department of Health and Human Services, Environmental Protection Agency, American Association of Blood Banks), was also reviewed and evaluated. Based on the findings from the available literature, the following sections of the manuscript provide a detailed review of healthcare supply chain management in the United States, focusing on the use of respirators, life-saving medical supplies, blood supply chains, disinfectants, and human remains handling and logistics.

#### 4.2.1. Use of Respirators

The guidelines of the Center for Disease Control and Prevention (CDC) of the United States are recommended for extended use as well as the limited reuse of N95 filtering facepiece respirators for healthcare settings. Based on the CDC guidelines [45], healthcare institutions are advised to (1) minimize the number of patients who need to use respirators through the preferential utilization of administrative and engineering controls; (2) consider alternative options to N95 respirators (e.g., powered air-purifying respirators, alternative types of filtering facepiece respirators); (3) rely on the practices that promote the extended use as well as the limited reuse of N95 respirators; and (4) prioritize the patients who require N95 respirators based on the associated complications due to infection. Healthcare institutions are also encouraged to refer to the emergency use authorization guidelines prepared by the U.S. Food and Drug Administration (FDA) for all disposable filtering facepiece respirators [46].

Moreover, the CDC developed a list of strategies for optimizing the supply of N95 respirators and for conducting other healthcare activities. These strategies can be categorized into three groups, including the following [47]: (a) conventional capacity strategies; (b) contingency capacity strategies; and (c) crisis capacity strategies. The conventional capacity strategies include the selective use of airborne infection isolation rooms, use of physical barriers, and properly maintained ventilation systems. Furthermore, there are some administrative controls under the contingency capacity strategies, such as limiting the number of patients going to hospitals, telemedicine, limiting face-to-face encounters with patients, excluding visitors from contact with potential patients with COVID-19, training on indications for the use of N95 respirators, just-in-time fit testing, and others. The contingency capacity strategies consist of decreasing the length of hospital stay for medically stable patients with COVID-19 and suspending the annual fit testing temporarily. The crisis capacity strategies include the following: (1) use of respirators beyond the design shelf life for healthcare delivery; (2) use of respirators approved under the standards that are used in other countries; (3) prioritize the use of respirators as well as facemasks by the type of activity; (4) use of ventilated headboards; and (5) use of expedient patient isolation rooms for risk reduction.

#### 4.2.2. Life-saving Medical Supplies

The Strategic National Stockpile is considered the nation’s largest supply of life-saving medical supplies and pharmaceuticals, especially for use during emergencies. The Strategic National Stockpile has various branches, including the following [48]: (a) information and planning branch; (b) management and business operations branch; (c) operational logistics branch; (d) science branch; and (e) strategic logistics branch. Effective operations of these branches are critical for the Strategic National Stockpile mission. The Strategic National Stockpile has been actively engaged during major national emergencies, including the Zika virus outbreak (2016–2017); Hurricanes Harvey, Irma, and Maria (2017); Hurricane Dorian (2019); and the COVID-19 pandemic (2020). However, local supplies of the Strategic National Stockpile are expected to reduce significantly due to the COVID-19 pandemic. As of April 2020, 11,055 tons of cargo were shipped by the Strategic National Stockpile to support the U.S. repatriation efforts, including 349 flights with the required supplies as well as 2218 trucks with the required supplies [48]. More than 175 staffers were assigned to serve the Stockpile’s operational center. Furthermore, over 50 private industry partners have been involved in medical supply and delivery.

#### 4.2.3. Blood Supply Chains

According to the Association for Health Care Resource and Materials Management (AHRMM) of the American Hospital Association, blood supplies were fairly tight before the occurrence of the COVID-19 pandemic [49]. COVID-19 is expected to negatively affect blood supplies even further. Supply chain professionals are recommended to collaborate with laboratory and clinical staff throughout the management and conservation of blood products, aiming to meet the demand for blood products for the remainder of the pandemic as well as after the pandemic. The following tips are recommended for hospitals: (1) use O-negative red blood cells (RBC) only for the patients with the anti-D antibody as well as women of childbearing age; (2) use A plasma specifically for massive transfusions; (3) develop effective strategies for situations with limited resources; (4) cross-match a unit of RBC for more than one patient; (5) use perioperative autologous blood salvage; (6) lower transfusion triggers for RBC and platelets; and (7) prioritize patients (e.g., emergency patients vs. non-emergency patients).

In March 2020, the American Association of Blood Banks (AABB) formed the AABB COVID-19 Working Group in order to support blood transfusion services and blood collection centers under the adverse impacts of the COVID-19 outbreak [50]. The AABB COVID-19 Working Group is expected to closely monitor the current COVID-19 situation and its effects on blood safety and availability, recruitment of blood donors, the safety of blood donors, and operational continuity. The AABB COVID-19 Working Group must also provide resources and guidelines for hospital transfusion services and blood centers to address the new challenges posed by COVID-19. Last but not least, the AABB COVID-19 Working Group is expected to closely work with the AABB Inter-organizational Task Force on Domestic Disasters and Acts of Terrorism, which is responsible for blood supply coordination during emergencies.

#### 4.2.4. Disinfectants

Individuals and healthcare institutions are encouraged to refer to the database of disinfectants that meet the criteria of the Environmental Protection Agency (EPA) of the United States for the use against SARS-CoV-2, which is considered as the cause of COVID-19. The EPA database provides the following information for each disinfectant [51]: (1) EPA registration number; (2) active ingredients; (3) product name; (4) production company; (5) disinfection directions; (6) contact time; (7) formulation type; (8) surface types for use; (9) use site; and (10) emerging viral pathogen claim. The CDC also provides detailed guidelines to individuals for disinfecting their households using EPA-approved disinfectants [52]. Generally, the CDC encourages individuals to follow particular instructions, which are available on the disinfectant label, to ensure the most effective and safe use of the product. Many disinfectants require keeping the surface wet for a certain period of time.

#### 4.2.5. Human Remains Handling and Logistics

The CDC has a set of guidelines for the collection and submission of postmortem specimens from the decedents that had COVID-19 infection [53]. Certain states developed their guidelines for human remains handling and logistics, most of which are based on CDC recommendations. For example, the Louisiana Department of Health (LDH) provides a detailed set of guidelines for funerals and handling human remains under the COVID-19 settings [54]. In particular, each funeral cannot exceed 10 people while the order is in place. Moreover, all cremations and funerals should be conducted within three days after the death date if there are no practical constraints. Funeral homes have to be notified in advance if the decedent that they will have to work with passed away due to COVID-19. Funeral home staff should follow the CDC guidelines for handling the decedent who passed away due to COVID-19.

Moving a recently deceased COVID-19 patient may still be hazardous for funeral home staff, as the lungs of the decedent may still expel a small amount of air. Funeral home staff are required to wear personal protective equipment (i.e., an impervious gown, N95 mask, eye protection, and double surgical gloves). Only essential personnel can be present in the room or morgue when handling the decedent. Particular steps are recommended before removing the decedent’s body, including the following [54]: (1) cover the decedent’s face entirely using a material soaked with a strong disinfectant; (2) ensure that the decedent’s identification is tagged to the body; (3) the entire body has to be wiped with a strong disinfectant when possible; (4) the body has to be placed in a designated lightweight body bag; (5) the body bag also has to be wiped with a disinfectant; (6) ensure that the decedent’s identification is tagged to the body bag; and (7) afterward the body has to be transported to the designated funeral home. After placing the decedent’s body in the vehicle, funeral home staff can remove personal protective equipment.

When a United States citizen dies outside the country, the decedent’s representative should notify the appropriate United States consular officials at the Department of State. An autopsy of a person who passed away overseas is not required by the CDC before returning the body to the United States. However, some countries may require an autopsy depending on the death circumstances. The human remains must meet the importation standards outlined in the Code of Federal Regulations, Part 71.55. Based on federal regulations, the import of human remains can be authorized if one of the following conditions is met [54]: (a) the remains are cremated; (b) the remains are placed in a hermetically sealed casket and properly embalmed; (c) the remains are accompanied by a permit that was issued by the CDC director.

## 5. Discussion

A concise summary of the conducted review of the healthcare supply chain management trends in Hong Kong and the United States under COVID-19 settings is presented in Table 1.

Based on the identified trends, the following strategies are recommended for the relevant stakeholders to improve healthcare supply chain management in Hong Kong and the United States under the COVID-19 settings:
Some of the strategies that are used in Hong Kong and the U.S. for optimizing the supply of N95 respirators and conducting other healthcare activities may seem harsh to a certain extent (e.g., the use of respirators beyond the design shelf life for healthcare delivery). Some additional alternatives should be considered in order to improve the existing strategies. In particular, countries that experience a sharp increase in COVID-19 cases should collaborate with countries that have surpassed the COVID-19 peak, so the countries that have a surplus of N95 respirators can share them with the countries that experience a shortage. Furthermore, there are some alternatives to extending the life of N95 respirators. It is possible to extend the use of N95 respirators for up to 8 h or reuse N95 respirators by following certain methods, such as mask rotation, decontamination/reprocessing, hydrogen peroxide vaporization, UV treatment, moist heat, and dry heat. Furthermore, the healthcare industry may collaborate with logistics firms and manufacturers to create a storage system for the extended use and reuse of N95 filtering facepiece respirators so as to stabilize the supply of N95 respirators in the long term [55]. In the near future, a new product (i.e., Aeri mask) is expected to be introduced to the healthcare market. This product can perform various functions (e.g., self-cleaning, facial recognition, antifogging, removable air filters, and incorporated fan for breathable relaxation) and overcome the shortage problem along with the shortcomings of N95 respirators [56].As the HA and the U.S. Strategic National Stockpile are expected to have a shortage of life-saving medical supplies and pharmaceuticals due to the COVID-19 pandemic, new policies should be developed to prioritize deliveries of life-saving medical supplies and pharmaceuticals among key locations (i.e., nursing homes, hospitals, healthcare facilities, and the regions that experience severe COVID-19 consequences should receive priority). In other words, healthcare supply chain stakeholders need to be responsive to the evidence instead of allocating the available resources on a first-come, first-served principle. As suggested by Lau et al. (2020) [4], an extensive distribution network can ensure the reliability of the inventory level for life-saving medical supplies and pharmaceuticals and the effectiveness of emergency material transportation. In order to solve the issue of deficiency of life-saving medical supplies and pharmaceuticals, expanding the resources to a minimum of two different countries and producing life-saving medical and pharmaceuticals in the home countries or cities rather than depending on imports would be viable alternatives [57].The U.S. American Association of Blood Banks (AABB) does provide some strategies for hospitals to tackle the issues that are associated with the blood supply shortage. However, these strategies have to be further tested and evaluated to determine their effectiveness for blood supply chains. On the other hand, the Hong Kong Red Cross is mainly concerned with the minimization of COVID-19 transmission via blood donation. This leads to shortages in the donated blood inventory level. The HA closely collaborates with the Hong Kong Red Cross to oversee emergency blood supplies and manage the donated blood inventory level. As expected, the COVID-19 pandemic and the termination of mobile blood drives further discourage donors from blood donations. As such, one viable alternative to address this issue would be the allocation of mobile blood drives traveling directly to donors’ homes following necessary COVID-19 precautions, especially in the case of lockdown and social distancing policies. Furthermore, blood conservation approaches, such as improved patient blood management, could also be adopted. Under patient blood management, the overall blood utilization could be enhanced. Additionally, better management of the demand for blood transfusions will help to protect blood stocks during the pandemic situation [58].The U.S. Environmental Protection Agency (EPA) provides a database for the recommended disinfectants against COVID-19. However, some regions of the country experienced a shortage of certain disinfectants (especially, during the COVID-19 peak period). A set of additional guidelines are needed for individuals and healthcare institutions on the use of alternative disinfectants in case of primary disinfectants being temporarily unavailable (due to the shortage issue). Similar to the United States, there is a wide range of disinfectants against COVID-19 in Hong Kong for the general public. To this end, the Hong Kong Consumer Council conducted a comprehensive investigation into the quality of disinfectants. Hong Kong residents are required to follow the Hong Kong Consumer Council recommendations on purchasing disinfectant products from reputable brands or renowned retailers. Interestingly, business firms, educational institutions, hospitals, and governments outsource disinfection services to professional service providers. Outsourcing the non-core business could help the firm minimize the investment risk, keep operational control, and concentrate on core tasks during the COVID-19 pandemic. Importantly, the firms may need to form a cross-disciplinary team to search for the best possible service providers and develop strategic partners. Both parties may jointly invest and produce local disinfectants so as to avoid relying on imported products.Both the United States (i.e., The U.S. Center for Disease Control and Prevention (CDC) and state Departments of Health) and Hong Kong (i.e., Food and Environmental Hygiene Department and Center for Health Protection (CHP)) have detailed guidelines for handling the remains of COVID-19 decedents. Additional tips based on the lessons learned throughout handling remains of the COVID-19 decedents (i.e., typical difficulties that were encountered in the past and how they can be effectively addressed) would be helpful for the stakeholders who are directly involved in human remains handling and logistics. Due to COVID-19, there is a remarkably increasing amount of human remains. However, flight schedules continuously change or are cancelled at a short notice due to the COVID-19 disruptions, which caused uncertainties in the delivery of human remains from one country to another. Human remains handling and logistics have seriously suffered. In order to address this issue, human remains logistics firms may form a partnership or sign an agreement with airlines to provide a charter flight or a specific flight in case of original flight changes. This will facilitate human remains logistics throughout the COVID-19 pandemic and post-pandemic periods.

Apart from the above-suggested measures, artificial intelligence (AI)-based methods could also be used to effectively address a wide array of different decision problems [59,60]. Ref. [61] conducted a bibliometric analysis of the AI-based studies and highlighted that healthcare supply chains in the AI context are considered an emerging topic; ref. [62] identified that healthcare and logistics experts strive towards employing AI and automation in response to the COVID-19 pandemic. In the digital era, AI technology can be adopted for preventive and predictive measures under COVID-19 settings. BlueDot is an illustrative example of adopting AI technology to forecast and oversee COVID-19 cases [63]. Ref. [63] further explains that “AI has proven to be a valuable tool in public health efforts, helping to characterize the epidemiology of COVID-19 and model disease transmission, even in the early stages of the pandemic”. The AI technology can be integrated with POC (Point of Care) diagnostics to facilitate the self-testing of patients for COVID-19; refs [64,65] used the AI system for predicting the deterioration of COVID-19 patients. Furthermore, ref. [66] applied the AI approach to analyze the severity of COVID-19. Such predictions could be used to prioritize COVID-19 patients for treatment. Moreover, ref. [58] employed the AI approach and the Geographic Information System (GIS) environment to effectively determine the virus spread, and ref. [67] forecast the economic recession through the logistics industry with artificial intelligence (AI) during the pandemic.

The AI-based approaches could serve as effective preventive measures, including temperature screening systems and social distancing. For example, ref. [68] discussed thermal infrared imaging-based screening. The system was employed for automatic face recognition and fever assessment. The screening system is generally used for mass screening at airports, hospitals, and schools. In the meantime, social control is one of the main measures to cease the spread of COVID-19. Ref. [69] proposed a social distancing detector using the convolution neural network (CNN), and ref. [70] proposed a system for the detection and tracking of people to implement social distancing measures by means of a novel deep learning detection technique. Such an AI-based approach could be applied to indoor and outdoor scenarios in real-time. Additionally, ref. [71] constructed an AI-based model that created a type of recurrent neural network (RNN) for epidemiological modelling and preventing the COVID-19 spread.

As a result of a detailed review of healthcare supply chain management in Hong Kong and the United States under COVID-19 settings, it was found that AI-based applications have still not been explored and implemented to their full potential. Application of the AI-based methods would assist healthcare managers with better utilization of the available healthcare resources, which will further minimize the risk and losses due to COVID-19 within a short period of time [63]. When it comes to preventive measures, AI should be more widely adopted, not only for thermal infrared image-based mass screenings and social distancing detection in crowded places (e.g., public transport and public activities), but also for more advanced preventive strategies (e.g., COVID-19 contact tracing applications) [8]. Furthermore, the AI-based methods would be effective for scheduling logistics operations (e.g., transportation of different medical supplies between hospitals serving COVID-19 patients, etc.) [72]. Apart from these, due to much of the traditional training being suspended during the pandemic, virtual reality (VR) technology was used in the local hospitals for training medical practitioners and patients [9,72]. The training enabled health workers to practice anytime and eliminated the need for face-to-face contact. In the future, healthcare and other relevant institutions should operate a flexible, adaptive, and resilient healthcare supply chain rather than resetting the whole healthcare system to deal with the challenges. AI will be one of the key tools that will foster achieving the Sustainable Development Goals (SDGs), notably those associated with giving general health coverage [73].

## 6. Conclusions

In our research study, we provided a detailed overview of healthcare supply chain management in Hong Kong and the United States under COVID-19 settings. To the best of our knowledge, this is the first comprehensive comparative research aimed at identifying the differences and common challenges imposed by COVID-19 on healthcare supply chains in Hong Kong and the United States. Hong Kong can be viewed as a representative country of Eastern regions of the world and is located close to Wuhan, China, where the pandemic started. On the other hand, the United States is one of the largest countries in the Western regions of the world, which experienced the largest number of COVID-19 cases and deaths as of June 2021. A detailed review of healthcare supply chain management under the COVID-19 settings in Hong Kong and the United States was conducted, focusing on some of the aspects that were found to be the most critical for both countries throughout the response to the pandemic, including the following: (a) use of respirators; (b) life-saving medical supplies; (c) blood supply chains; (d) disinfectants; and (e) human remains handling and logistics.

The outbreak of the COVID-19 pandemic greatly disrupted the healthcare supply chain across the world. Sustainability and resilience need to be identified as vital attributes of every healthcare supply chain. Hence, healthcare supply chain managers need to have a plan to enhance the sustainability and resilience of their healthcare supply chains. These healthcare supply chains would effectively react to the dynamic context under the pandemic settings. Insufficient sustainable and resilient healthcare supply chains would increase the impacts of a pandemic and generate extra negative outcomes. Additionally, innovative technologies such as AI and blockchain are valuable vehicles that could be adopted in managing healthcare supply chains in the event of healthcare supply chain catastrophic disruptions. Blockchain can improve the traceability and transparency of healthcare supply chains, while AI can be deployed to assess the effectiveness of various strategies and implement response plans by conducting optimization algorithms. In addition, the government, healthcare managers, and policymakers may develop a strategic plan to deal with a catastrophic healthcare supply chain disruption. Most of the existing plans are now only focusing on either response or preparation stages, which may not be the best approaches or the most effective way to mitigate the effects of COVID-19 on healthcare supply chain operations. Thus, healthcare supply chain stakeholders may prioritize the planning of response, preparation, and recovery activities so as to make better alignment with key healthcare supply chain disruptions in the forthcoming years. In terms of preparedness, typical activities include intertwined supply, diversifying sourcing, computerized modelling, local manufacturing, multiple sourcing, online delivery, and vertical planning. Responses include illustrative activities pertaining to communication, flexible (situational) response, distributed manufacturing, additive manufacturing, diversifying demand, increasing production, prioritizing demand, regulating purchases, information sharing, local supply, and equipment reuse. Recovery includes representative activities such as government regulation, government support, and tax cuts [9].

Some common challenges were identified for healthcare facilities in Hong Kong and the United States. In particular, due to shortages of N95 respirators, fairly harsh strategies were adopted in both countries (e.g., the use of respirators beyond the design shelf life for healthcare delivery). Export bans further restricted supply to importing countries. Along with shortages in N95 respirators, the pandemic caused shortages in life-saving medical supplies and pharmaceuticals. To the best of the authors’ knowledge, there is a lack of workers to transport and produce healthcare products due to workers not showing up to work or being sick [74,75]. To a certain extent, the existing healthcare supply chain shows a lack of visibility. Poor visibility arises from insufficient rapid access to real-time, centralized, and consumable data from comprehensive systems and dispersed data sources. This leads to it being extremely difficult to decide what is in stock, what the replenishment level is, and the range of future demand. Eventually, industrial practitioners are unable to manage, predict, and measure in response to panic buying. Additionally, poor integration and alignment lead to discouraging an organization to collaborate and a fragmented method of fulfillment and order during the COVID-19 pandemic. Blood supply chains experienced a lack of blood donors during the COVID-19 pandemic along with additional restrictions for potential blood donors based on the newly established COVID-19 protocols. COVID-19 has highlighted a need for improving overall resiliency in the healthcare supply chain. One possible method to enhance supply chain resiliency is by adding redundancy in the healthcare supply chain, for instance by contracting backup suppliers, holding extra manufacturing capacity for creating critical items, diversifying the supply base, and carrying additional stock of crucial healthcare items. The public health agencies in Hong Kong and the United States provided detailed guidelines to the public on the use of different disinfectants, aiming to prevent the virus spread. Furthermore, additional measures were introduced for handling the remains of COVID-19 decedents. A set of potential strategies were proposed in this study that could be implemented to address the aforementioned challenges. Currently, there is a lack of innovation, technology solutions, and high-tech manufacturing involved in the healthcare supply chain. The potential benefits of using AI-based methods in healthcare supply chain management under COVID-19 settings may integrate data across the whole healthcare supply chain and enhance supplier monitoring. As expected, it may deal with a global emergency and build integrated healthcare supply chains between logistics operators, manufacturers, suppliers, and hospitals [9,14].

The research topic is interesting and may prove useful to researchers, health professionals, policymakers, local communities, and government bodies. However, the study has certain research limitations that could be considered as future research directions and agendas. In future studies, a set of interviews could be conducted with healthcare professionals in Hong Kong and the United States, aiming to determine the healthcare practices that were found to be efficient and to serve the public in the COVID-19 settings. These healthcare practices could be further enhanced and implemented in response to the pandemics that may occur in the following years. As such, it may help to fill the research gap in the field of disaster management and emergency logistics. Second, the perception of healthcare professionals towards the application of different AI-based methods could be investigated and assessed further. Technical sustainability of cloud-based blockchain could be combined with AI for healthcare supply chain management [76]. Third, some companies intend to produce new types of personal protective equipment (PPE) so as to develop a new business and sustain public health services. To this end, more research is needed to assess the advantages of the new PPE types as well as the potential challenges in their use. Fourth, future research studies focusing on a more detailed evaluation are needed for different approaches that can be used to address the shortages in life-saving medical supplies and pharmaceuticals. This will assist healthcare professionals in the prevention of potential shortages due to various healthcare supply chain disruptions (including future pandemics). Fifth, the healthcare supply chain may design and implement a decision analysis system to rank the main alternatives/criteria. A decision analysis system may optimize operational efficiency and develop a lean system to minimize waste.

## Figures and Tables

**Figure 1 healthcare-10-01549-f001:**
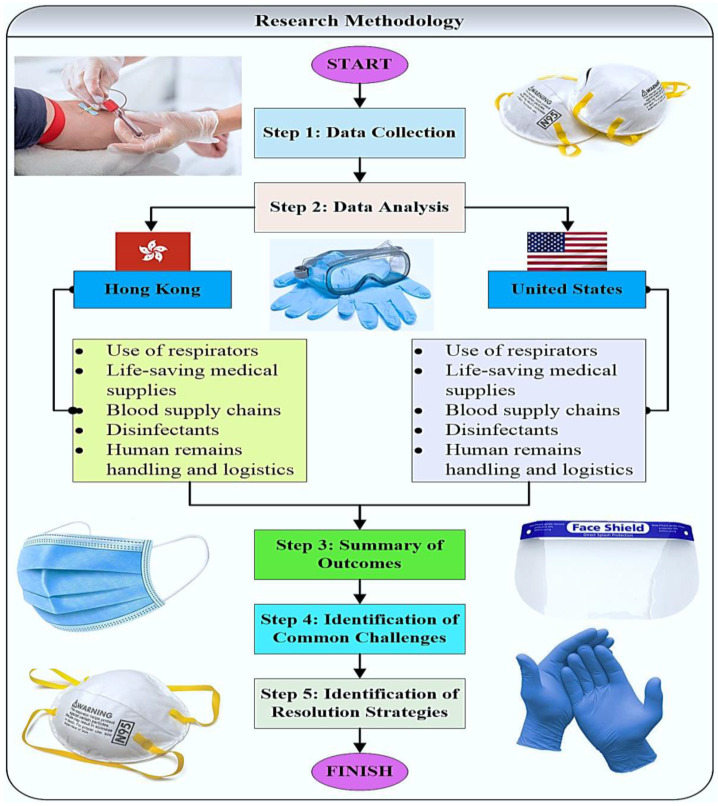
The mains steps of the adopted research methodology.

**Table 1 healthcare-10-01549-t001:** Summary of the healthcare supply chain management trends in Hong Kong and the United States under the COVID-19 settings.

Attribute\Country	Hong Kong	United States
Use of respirators	Follow the CHP standards Encourage staff to change the protective gear, such as N95 respirators, less frequently Ensure that N95 respirators are available for the frontline healthcare professionals	Minimize the number of patients who need to use respirators through the preferential utilization of administrative and engineering controls Consider alternative options to N95 respirators Prioritize the patients who require N95 respirators
Life-saving medical supplies	As of August 2020, Hong Kong hospitals received around 2900 cubic meters of medical supplies and PPE The HA collaborates with Cathay Pacific Cargo to address the shortage in life-saving medical supplies inventory control and provide flexible service delivery PPE is for aerosol-generating procedures (AGPs)	As of April 2020, 11,055 tons of cargo shipped by the Strategic National Stockpile to support the U.S. repatriation efforts Life-saving medical supplies are expected to reduce significantly due to the COVID-19 pandemic Private industry partners have been involved in medical supply and delivery
Blood supply chains	Defer for 28 days from blood donation for those who recently visited a place with active community transmission of COVID-19 Defer for 28 days from blood donation for those who had close contact with a confirmed case of COVID-19 Defer for 180 days from blood donation for those who tested positive for COVID-19	The COVID-19 pandemic is expected to cause shortages in blood supplies Supply chain professionals are recommended to collaborate with the laboratory and clinical staff to meet the demand for blood products A number of tips were introduced to effectively manage blood products
Disinfectants	Research institutions and universities identified some disinfectants that are suitable for skin, while some disinfectants are appropriate for surfaces according to the duration of exposure and the concentration level The government suggested the general public buy disinfectants with detailed product information of reputable brands from renowned retailers The disinfection service providers need to comply with the EPD and CHP	Individuals and healthcare institutions are encouraged to refer to the database of disinfectants that meet the EPA criteria The EPA database provides more details for each disinfectant The CDC also provides detailed guidelines to individuals for disinfecting their households
Human remains handling and logistics	Category 2 of human remains handling, which allows cremation and disallows embalming A full PPE protection with appropriate resuscitation rooms Robust and leak-proof transparent plastic bags not less than 150 μm thick and zippered closed should be used Bodybag should be wiped in 1-in-4 diluted household bleach and allowed to air dry Bodybag should be identified by a yellow label/tag	The CDC has a set of guidelines for the collection and submission of postmortem specimens from the decedents that had COVID-19 infection Certain states developed their guidelines for human remains handling and logistics, most of which are based on the CDC recommendations Funeral home staff should follow the CDC guidelines for handling the decedent who passed away due to COVID-19 A full PPE protection is required for the funeral home staff

## Data Availability

Not applicable.

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
