# Peer review of "Healthcare Supply Chain Management under COVID-19 Settings: The Existing Practices in Hong Kong and the United States"

_healthcare, 2022, doi:10.3390/healthcare10081549_

Round 1

Reviewer 1 Report

The paper is interesting. There are two minor issues (the second one is a suggestion) to improve the paper.

Some matters that have been discussed in this paper are similar to providing services, not supply chain management. I suggest that the authors provide a comprehensive insight into supply chain management in healthcare, see Sibevei, Ali, et al. "Developing a Risk Reduction Support System for Health System in Iran: A Case Study in Blood Supply Chain Management." International journal of environmental research and public health 19.4 (2022): 2139.. In addition, I suggest that the authors first bring a good classification for the topics that they are discussing in this paper. They are explored very arbitrarily. I suggest the author first bring a big picture and then discuss each component one by one.

In addition, the mathematical model and approaches are widely used in this research area, see Moosavi, Javid, Amir M. Fathollahi-Fard, and Maxim A. Dulebenets. "Supply chain disruption during the COVID-19 pandemic: Recognizing potential disruption management strategies." International Journal of Disaster Risk Reduction (2022): 102983.. I recommend that the authors add a discussion regarding this matter to the paper.

Reviewer 2 Report

Manuscript ID: healthcare-1841279

Title: Healthcare supply chain management under the COVID-19 Settings: The existing practices in Hong Kong and the United States

This topic is interesting and seems useful to the readers. However, there are some concerns to be resolved at this stage based on the following comments:

1- Please improve the English and writing of the manuscript. I found several errors; for example, replace “lack information” with “a lack of information”.

2- Please be consistent about the application of uppercase and lowercase letters to spell out the acronyms.

3- The main findings of the research are not clear. You should go through the achievements by previous studies and find the opportunities to improve the healthcare systems based on the experiences.

4- What are the best approaches for healthcare SCM in these two countries?

5- The challenges should be discussed in more detail.

6- It would be better if you could consider a decision analysis system to rank the main alternatives/criteria; or you can look at it as a potential future research.

Reviewer 3 Report

1-Please, motivate more the abstract, trying to be more concise. Why this work is necessary?

2-Please revise the keywords and choose realistic ones.

3-Some sentences need reference for instance:

Healthcare supply chain operations have been 71 substantially affected by COVID-19. A number of major healthcare facilities (e.g., distribution centers, factories, and warehouses) are currently situated in COVID-19’s quarantine areas in China, Italy, India, the United States, Japan, Russia, and Australia.

4. Improve the literature review. Also you need to look for recent studies and remove those that are more than 5 years, unless they are important. It is required that you look to further studies in high impact factor journals and limit to large extent those form conferences. You can use the following in the field of COVID-19

 (2022). A sustainable-resilience healthcare network for handling COVID-19 pandemic. Annals of Operations Research312(2), 761-825.

(2021). Simulation-based optimization: analysis of the emergency department resources under COVID-19 conditions. International journal of industrial and systems engineering1(1), 1504.

5-You have ignored many features in the field of Blood Supply Chains in COVID-19 please see:

(2021). A stochastic bi-objective simulation–optimization model for plasma supply chain in case of COVID-19 outbreak. Applied Soft Computing112, 107725.

6-Replace ArXiv papers (if any) (unless very related to the article research area) with related articles from high impact factor journals. These articles from arXiv are not reviewed and therefore you must look for reviewed journal related work from www.sciencedirect.com

7- The assumptions are not clearly justified.

8-It would be better to arrange the paper based on: introduction, literature view, methodology, case study, results and conclusion

9-Research limitations can help researchers overcome obstacles. Please add research limitations to the conclusion section.

10-It is suggested that implication for managers be described in a paragraph of the conclusion section. In this paragraph, it should be explained that how research outputs help relevant managers and readers.

11- The writing of the paper needs a lot of improvement in terms of grammar, spelling, and presentations. The paper needs careful English polishing since there are many typos and poorly written sentences.
Some examples are as the following:

*Check the usage of the commas carefully.

* Check the articles including "a", "an" and "the".

* Check the required and unneeded blank spaces.

Round 2

Reviewer 2 Report

Good work. It can be now accepted.

Reviewer 3 Report

No comment